# Phylointeractomics reconstructs functional evolution of protein binding

Dennis Kappei[1,2,*], Marion Scheibe[3,4,*], Maciej Paszkowski-Rogacz[2], Alina Bluhm[3], Toni Ingolf Gossmann[5], Sabrina Dietz[3], Mario Dejung[3], Holger Herlyn[6], Frank Buchholz[2,7,8,9,10], Matthias Mann[4] & Falk Butter[3]

Molecular phylogenomics investigates evolutionary relationships based on genomic data. However, despite genomic sequence conservation, changes in protein interactions can occur relatively rapidly and may cause strong functional diversification. To investigate such functional evolution, we here combine phylogenomics with interaction proteomics. We develop this concept by investigating the molecular evolution of the shelterin complex, which protects telomeres, across 16 vertebrate species from zebrafish to humans covering 450 million years of evolution. Our phylointeractomics screen discovers previously unknown telomere-associated proteins and reveals how homologous proteins undergo functional evolution. For instance, we show that TERF1 evolved as a telomere-binding protein in the common stem lineage of marsupial and placental mammals. Phylointeractomics is a versatile and scalable approach to investigate evolutionary changes in protein function and thus can provide experimental evidence for phylogenomic relationships.

[1] Cancer Science Institute of Singapore, National University of Singapore, 14 Medical Drive, Singapore 117599, Singapore. [2] Medical Systems Biology, UCC, University Hospital and Medical Faculty Carl Gustav Carus, TU Dresden, Fetscherstraße 74, Dresden D-01307, Germany. [3] Institute of Molecular Biology (IMB) gGmbH, Ackermannweg 4, Mainz D-55128, Germany. [4] Department of Proteomics and Signal Transduction, Max Planck Institute of Biochemistry, Am Klopferspitz 18, Martinsried D-82152, Germany. [5] Department of Animal & Plant Sciences, University of Sheffield, Western Bank, Sheffield S10 2TN, UK. [6] Institute of Anthropology, University of Mainz, Anselm-Franz-von-Bentzel-Weg 7, Mainz D-55099, Germany. [7] Max Planck Institute of Molecular Cell Biology and Genetics, Pfotenhauerstraße 108, Dresden D-01307, Germany. [8] German Cancer Research Center (DKFZ), Neuenheimer Feld 280, 69120 Heidelberg, Germany. [9] German Cancer Consortium (DKTK) partner site, Fetscherstr. 74, 01307 Dresden Germany. [10] National Center for Tumor Diseases (NCT), University Hospital Carl Gustav Carus, TU Dresden, Dresden D-01307, Germany. * These authors contributed equally to this work. Correspondence and requests for materials should be addressed to F.Buchholz (email: frank.buchholz@tu-dresden.de) or to M.M. (email: mmann@biochem.mpg.de) or to F.Butter (email: f.butter@imb-mainz.de).

The analysis of evolutionary relationships of gene sequences was strongly advanced in recent years because of the advent of high-throughput DNA sequencing technologies[1,2]. However, while overall gene sequence might be conserved, even single amino-acid exchanges can change the functionality of the corresponding proteins and thus drives evolution in regulatory networks[3,4]. Mass spectrometry-based interaction proteomics has been the technique of choice to identify protein interactions in a systematic manner[5].

To systematically assess functional evolutionary changes in protein binding, we developed the concept of phylointeractomics combining phylogenomics with interaction proteomics. We demonstrate the power of our approach by investigation of telomere-binding proteins across 16 vertebrate species, sharing the common telomeric repeat motif TTAGGG[6]. These repeats are bound by the shelterin complex, which protects the linear chromosome ends from recognition as DNA double-strand breaks and is composed of six subunits (TERF1, TERF2, TIN2, TPP1, RAP1 and POT1) in human. It is generally assumed that throughout 450 million years of vertebrate evolution[7] this complex is conserved[8,9]. However, experimental validation of this phylogenomic assumption is lacking.

Here we show that TERF1 actually evolved its intrinsic telomere-binding ability in the common stem lineage of marsupial and placental mammals. This observation exemplifies that the assumption to equate phylogenomic homology and functional conservation has restrictions, and with phylo-interactomics, we provide a versatile and scalable approach to uncover these functional differences.

## Results

**Interaction proteomics recapitulates the core telosome**. In our phylointeractomics screen to identify telomere-binding proteins, we used a DNA-protein interaction approach combined with quantitative mass spectrometry. Polymerized biotinylated DNA of either telomeric sequence (TTAGGG) or a scrambled control sequence (GTGAGT) was immobilized on paramagnetic streptavidin beads. Both sequences were separately incubated with nuclear protein extracts from each of the 16 species (Fig. 1a), and bound proteins were analysed by label-free quantitative mass spectrometry (Fig. 1b). We discovered significant interactors (identical cutoff values of $S_0 = 0.6$ and $P < 0.05$, see Methods section) with the telomeric TTAGGG sequence by quadruplicate pull-down experiments for all 16 vertebrates. Each pull-down experiment quantified on average 1,300 proteins. This allowed us to determine significant interactors by their enrichment rather than the presence or absence on target and control sequence, making the analysis much more robust[10]. To further focus on high confidence interactors in the core telosome that potentially have a role in a larger number of organisms, we focussed on those candidates that were identified in at least five species (Figs 1c and 2a, Supplementary Data 1 and 2).

The shelterin complex (TERF1, TERF2, TIN2, TPP1, RAP1 and POT1) involved in telomere end protection and telomere homeostasis clustered tightly together with high enrichment scores in mammals, providing a positive control (Figs 1c and 2a, Supplementary Data 1 and 2). Furthermore, we observed the recent gene duplication of POT1 into POT1A and POT1B in the rodent lineage[11] when using extracts from mouse and rat and we identified both TERF2 paralogues in medaka. This demonstrates the required comprehensiveness and sensitivity to detect specific molecular evolution events (Fig. 1c).

**Phylointeractomics identifies putative novel telomere binders**. Our screen recapitulated the TERF2-interaction partner DCLRE1B, a nuclease implicated in proper end processing[12,13], and the nuclear receptors NR2C2 and NR2C1, previously described as subtelomere-binding proteins[14–16]. So far, only the homeobox-domain-containing proteins TERF1, TERF2 and HOT1 have been demonstrated to directly bind to double-stranded telomeric DNA while both POT1 and the CST complex are single-stranded telomeric binders[17–19]. In addition to the established double-strand binders, we identified eight zinc finger proteins (ZBTB7A, ZBTB10, ZBTB48, ZNF276, ZNF524, ZNF827, VEZF1 and KLF12) enriched at the TTAGGG repeat sequence in at least five species. We speculate that some of these candidates bind double-stranded TTAGGG repeats directly and have functional roles at the telomere. For instance, human ZNF827 has, in the meantime, been reported to localize to telomeres, to induce telomere remodeling and to promote telomere–telomere recombination[20]. Consistently, with our observed enrichment of RECQL1 to TTAGGG repeats in 8 species, this helicase is involved in telomere maintenance, actively resolving telomeric D-loops and Holliday junction substrates, regulated via an interaction with TERF2 in human cells[21]. The recently discovered direct telomere-binding protein HOT1 was also enriched on telomeric DNA; consistent with a role of this protein in active telomere elongation, HOT1 is identified predominantly in those species with detectable telomerase activity (Fig. 2a, Supplementary Fig. 2) even in normal somatic tissue[22] as used in this screen, which is in agreement to its previously described differential binding behaviour in human cells, in which it associates with telomeres in cellular contexts with active telomere elongation[17]. Indeed, when performing the same analysis with nuclear protein extracts from telomerase-positive HeLa cells we readily detect HOT1 among the specific telomere binders, but it is not identified from the telomerase-negative IMR90 human fibroblasts used in our initial screen (Fig. 2a, Supplementary Fig 1a, Supplementary Fig. 2). RUNX1, RUNX2, CBFB and, in fewer cases, also RUNX3 (Fig. 2a, Supplementary Data 1 and 2) as well as three poly r(C) binding proteins PCBP1, PCBP2 and PCBP3 were consistently enriched at telomeric DNA in various vertebrates, including humans. CBFB had been previously detected by proteomics of isolated chromatin segments (PICh)[15] and PCBP1 as shelterin-associated by immunoprecipitation/mass spectrometry (IP/MS)[23]. RUNX proteins are transcription factors and RUNX1 regulates the differentiation of hematopoietic stem cells, whereas the PCBP proteins are generally thought to be RNA-binding proteins regulating several cancer relevant transcripts[24,25]. Of note, two other candidates, FSBP and NAIF1, contain a homeobox/myb domain, typical for double-strand telomere binders[17,26], and SRBD1 features an OB-fold as found in POT1 (ref. 18), suggesting that these candidates may bind directly to TTAGGG and regulate and maintain telomeres. Thus our screen resulted in numerous proteins already implicated in telomere regulation and provides evidence for several novel candidates.

The affinity purification assay[27] used in this study may have putative limitations that need to be carefully considered. The assay primarily identifies direct binding proteins to a particular DNA sequence, together with their tight interaction partners. This is highlighted by the fact that we have identified the entire shelterin complex together with DCLRE1B but none of the more transient shelterin interactors[18] (Fig. 2a, Supplementary Fig. 1, Supplementary Data 1 and 2). Thus, to address research questions that involve temporal and spatial resolution at telomeres other interactomics approaches such as PICh[15] or a quantitative telomeric chromatin isolation protocol (QTIP)[28], an adaptation of the more general concept of combining chromatin immunoprecipitation with MS[5,29,30], have been developed. However, quantitative telomeric chromatin isolation

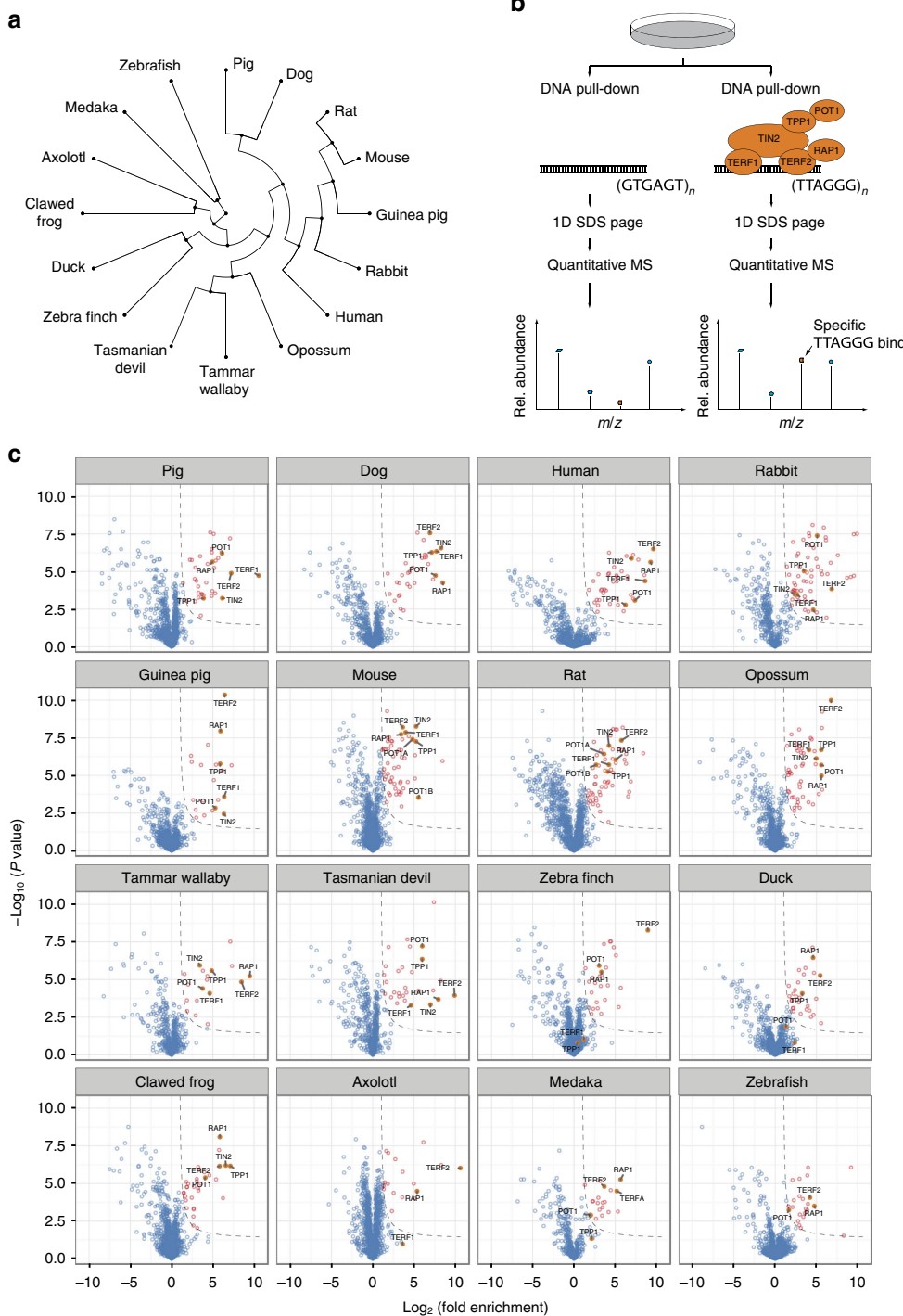

**Figure 1 | Phylointeractomics screen for telomere binders in 16 vertebrate species.** (**a**) Phylogenetic tree of vertebrate species analysed in this study—all higher ranked vertebrate taxa apart from monotremata are represented. (**b**) Quantitative label-free DNA interaction screen with DNA oligonucleotides containing either the telomeric repeat sequence (TTAGGG) or a control sequence (GTGAGT). Specific interaction partners are differentiated from background binders by a ratio significantly different from 1:1. All pull-downs were performed in biological replicates ($n = 4$). (**c**) Volcano plots for all tested vertebrate species. Specifically enriched proteins (red circles) are distinguished from background binders (blue circles) by a two-dimensional cutoff with $S_0 = 0.6$ and $P < 0.05$ (Welch's $t$-test). Detected members of the shelterin complex (TERF1, TERF2, TIN2, TPP1, RAP1 and POT1) are highlighted (filled orange dots).

protocol is specifically targeted to telomeres and dependent on the behaviour of TERF1 and TERF2, while PICh is currently limited to repetitive elements that provide multiple binding sites for sufficient enrichment of endogenous chromatin. These technical biases are likely the reason why other studies aiming

at identifying novel telomeric factors are showing surprisingly little overlap between each other beyond the shelterin proteins (Supplementary Fig. 3). Interestingly, when compared with other screens, our approach shows the strongest overlap with PICh, which uses telomeric DNA as the bait as well. Affinity purification

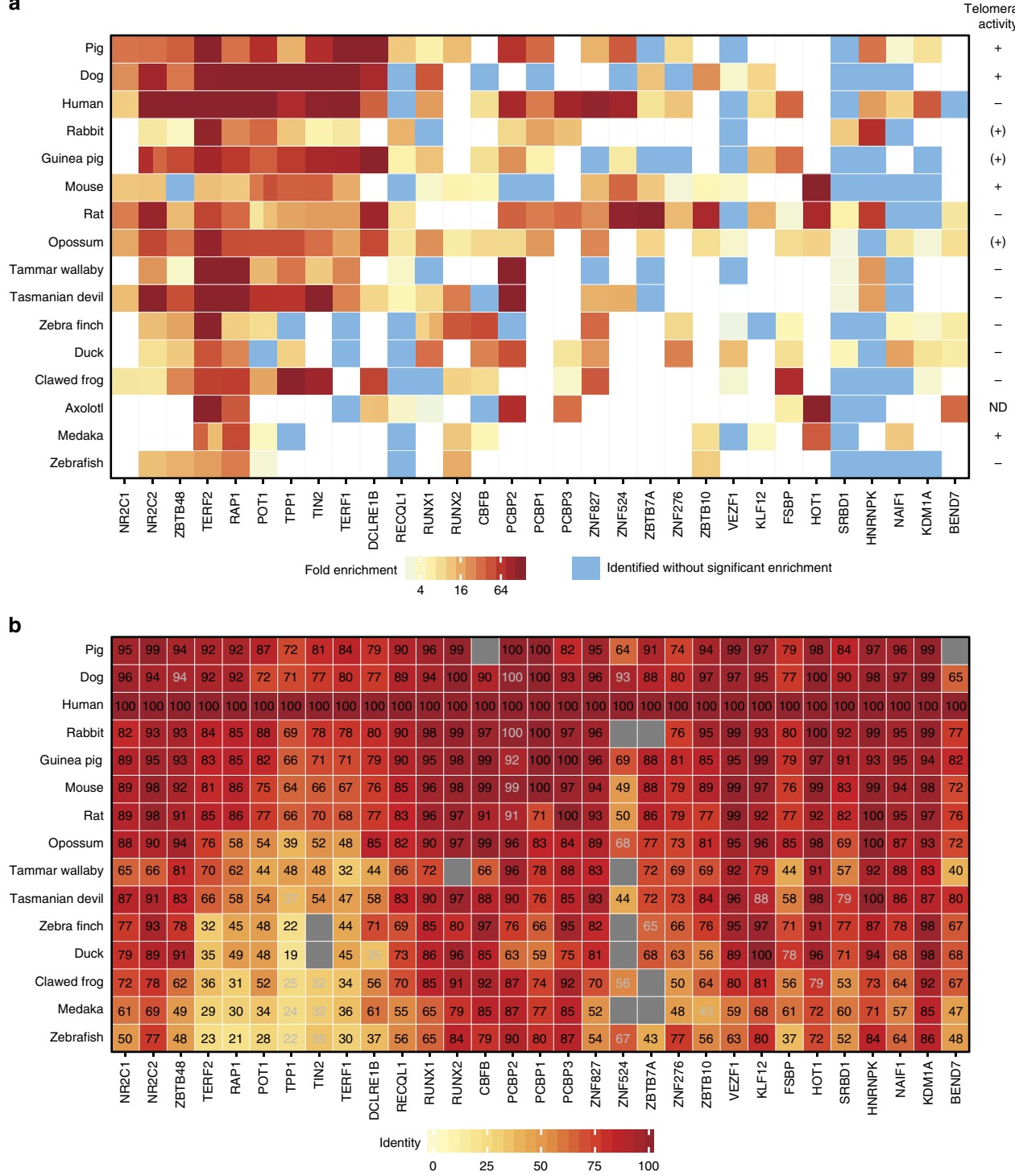

**Figure 2 | Telomere-binding proteins are identified by cross-species validation.** (**a**) Heat map of identified proteins with specific enrichment at TTAGGG repeats in at least 5 out of the 16 vertebrate species. Species (rows) are arranged according to their phylogenetic relations while proteins (columns) are clustered based on their binding pattern across all pull-downs. Colour gradient represents relative enrichment from TTAGGG binding (red) to equal enrichment on the telomeric and control sequence (blue). Only events that were identified as hits according to the criteria in Fig. 1c are shown. Gene names are based on the human versions, and in the occurrence of paralogues, both enrichment values are displayed side by side in the species concerned (for example, POT1 in mouse). The presence of telomerase activity is indicated on the right. ( + ) indicates cell lines with activity at the detection limit. IMR90 (human)[47] and NIH3T3 cells (mouse)[48] are known telomerase-negative and -positive cells, respectively. Quantifications can be found in Supplementary Fig. 2. (**b**) Heat map of protein sequence identity for all candidate telomere binders from **a** across 15 vertebrate species. Axolotl was excluded from this analysis as there is currently no published whole-genome annotation available. The protein sequence identity is displayed as a colour gradient relative to the human sequences with the percentage value displayed in each square. Grey squares represent absent genes based on ENSEMBL genome assemblies. Grey values represent non-annotated homologues based on ENSEMBL genome assemblies for which we could identify homologues by reciprocal BLAST search.

procedures such as the widely used immunoprecipitations and peptide pull-downs are prone to reveal all putative interaction candidates. Therefore, a careful validation of localization, binding and functional relevance in an endogenous context is required for any novel factor. Nevertheless, the presence of several already established telomeric factors in our screen suggests that our candidates may be important for telomere biology pending further in-depth characterization for each protein.

**Constitutive TERF1 telomere binding is a therian invention.** Importantly, our phylointeractomics approach also provides information about the evolutionary history of binding patterns of individual proteins, which here led to unexpected findings. Although the shelterin complex is thought to be universal to vertebrates[8,9] and indeed TERF2 and RAP1 are strongly enriched on TTAGGG repeats in all species analysed, TERF1 was absent or not specifically enriched in non-therian vertebrates despite the presence of a TERF1 orthologue in all 16 species (Figs 1c and 2a,b). Our data therefore raised the possibility that only therian (placental and marsupial) TERF1 can bind to telomeric TTAGGG repeats. To investigate this hypothesis, we recombinantly expressed the TERF1 DNA-binding domain (DBD) of 13 vertebrate species spanning from ray-finned fish to human and performed DNA-binding assays. Although all 8 recombinantly expressed TERF1 homeobox domains of therian mammals clearly showed binding to TTAGGG repeats, no binding was detectable for any TERF1 DBD outside this group (Fig. 3b). Our data therefore places a gain of direct TTAGGG repeat binding of TERF1 in the therian lineage after separation from monotremata around 225 million years ago.

**Specific TERF1 residues were positively selected in therians.** We next performed domain-specific multiple sequence alignment and conducted substitution rate analysis to infer selective pressures between therians and other species sampled. Using a branch-site model, selective pressures across sites were inferred from posterior distributions using a Naive Empirical Bayes (NEB) approach (Fig. 3a,c). We pinpointed seven amino-acid positions (A371, W374, Q378, K379, S382, Q393, S397 based on the opossum sequence) in the TERF1 DBD that are positively selected in therians but that evolve neutrally across non-therian branches. W374 and S382 are identical between platypus and therians and therefore cannot explain the observed binding pattern. Of the remaining five residues, only the Q393 residue confers direct contact with DNA based on the co-crystal structure of the human TERF1 homeobox domain with telomeric DNA[26]. Two additional residues, T400 and E420, differ between platypus and theria but are not under evolutionary constraints based on our branch-site model (Fig. 3a).

To test whether the identified amino-acid residues are important for telomeric binding, we mutated all residues that differ between platypus and therians individually and performed DNA-binding assays with the recombinant expressed opossum TERF1 DBD variants. In this screen, A371M, T400Q and E420D did not show any obvious change in their binding behaviour while Q378M, K379Q and Q393T reduced TERF1 binding. Exchange of S397 against the bulkier phenylalanine residue completely abrogated binding to telomeric DNA (Fig. 3d). Based on this information, we attempted to recapitulate a gain-of-function-binding switch for the platypus TERF1 DBD. To further quantify the change in binding activity, we used purified recombinant TERF1 domains and tested multiple platypus variants along with the platypus and opossum wild-type DBDs. Although the exchange of the bulky phenylalanine in F313S was not sufficient to induce binding, the combination with other

substitutions (T309Q and M294Q/Q295K) transferred TTAGGG binding to the TERF1 DBD from platypus (Fig. 3e,f, Supplementary Fig. 4), demonstrating that these are indeed the key residues for the evolutionary switch of TERF1 telomere binding.

## Discussion

Although TERF1 has not been extensively studied in non-therian vertebrates, available data from clawed frog and chicken suggests that TERF1 binding to telomeres and its telomeric function might indeed not be conserved. Using *in vitro* transcribed/translated *Xenopus laevis* TERF1, binding to chromatin or a plasmid containing TTAGGG repeats could only be observed when added to mitotic but not to interphase extracts with the mitotic phenotype appearing rather faint compared with that of TERF2 (ref. 31). Although direct binding to TTAGGG was not assessed, the fact that a specific extract has to be present to achieve any enrichment suggests that the interaction is likely indirect. For chicken TERF1 (cTERF1), Okamoto *et al.*[32] have reported as data not shown that cTERF1 can localize to telomeres in chicken cells. However, FLAG-cTERF1 does not localize to telomeres in mouse cells but telomeric localization can be enforced by fusing the chicken TRFH domain of TERF1 to the mouse TERF1 DBD[32]. Similarly, Cooley *et al.*[33] reported the localization of myc-cTERF1 to telomeres in DT40 *Terf1*$^{-/-}$ cells[33]. As the result by Okamoto *et al.*[32] could suggest indirect recruitment of cTERF1 to telomeres in an endogenous context, we performed a TTAGGG pull-down using nuclear protein extracts from 6C2 chicken cells (Supplementary Fig. 1b). In contrast to the results obtained with zebra finch and duck, cTERF1 is specifically enriched on telomeric DNA in our proteomics analysis (Supplementary Fig. 1b). Given that the cTERF1-DBD does not directly bind to telomeric DNA *in vitro* (Fig. 3b), non-therian TERF1 seems to be passively recruited to telomeres in some species and/or specific cellular contexts (for example, chicken, mitotic cells in xenopus). Importantly, our findings suggest that among the investigated species only therian TERF1 DBDs have an intrinsic TTAGGG-binding ability. A difference in cellular function is also supported by the observation that DT40 *Terf1*$^{-/-}$ cells do not display any defective cell viability contrary to mouse *Terf1* knockout cells[34] or major telomere dysfunctions[33]. These data functionally underscore our finding and indeed Cooley *et al.*[33] suggest as one possible explanation that the shelterin complex composition in chicken may differ from mammals.

Our results raise interesting questions about the evolution of the shelterin complex in vertebrates. Although TERF1 directly binds to telomeric dsDNA in therians, it is also connected to the shelterin complex via direct interaction with TIN2, mediated by the TRFH domain of TERF1 and the FxLxP motif in the C-terminus of TIN2 (ref. 35). Furthermore, TIN2 connects the remaining shelterin members through direct interactions with both TERF2-RAP1 and TPP1-POT1. Similar to TERF1, TIN2 and TPP1 are absent or not enriched in our pull-downs from several non-mammalian vertebrates (Figs 1c and 2a). Absence of TIN2 in interaction screens with bird proteomes turned out to be due to a lack of an annotated *TIN2* gene in this lineage in current genome assemblies (Fig. 2b). Without the central TIN2 hub, the six protein complex does not exist in birds. In the remaining non-mammalian vertebrates, the complex stability seems also impaired by the lack of TERF1 or at least by the lack of direct TERF1-TTAGGG binding (Fig. 1c). In addition, branch-site modeling of the TERF1 TRFH domain, which is important for dimerization[36], reveals the opposite conservation pattern compared with the homeobox domain as several residues are under purifying selection in non-therian vertebrates and evolve

neutrally otherwise (Supplementary Fig. 5). For both the homeobox and the TRFH domain in TERF2, which together with its interactor RAP1 is consistently bound to TTAGGG in all the 16 vertebrates in our screen, we could not detect such evolutionary differences (Supplementary Table 1 and 2). Together our data suggest that, after gene duplication of the ancestral TERF gene, TERF2 retained telomere-binding activity, whereas TERF1

evolved otherwise. Although it is possible that convergent evolution of TERF1 telomere binding is yet to be discovered in specific lineages of non-therian vertebrates, our data clearly illustrate that TERF1 regained the ability to directly bind TTAGGG repeats in therians where the six protein shelterin complex is found in its previously described composition and function[18]. Beyond the specific example of telomeres, these data

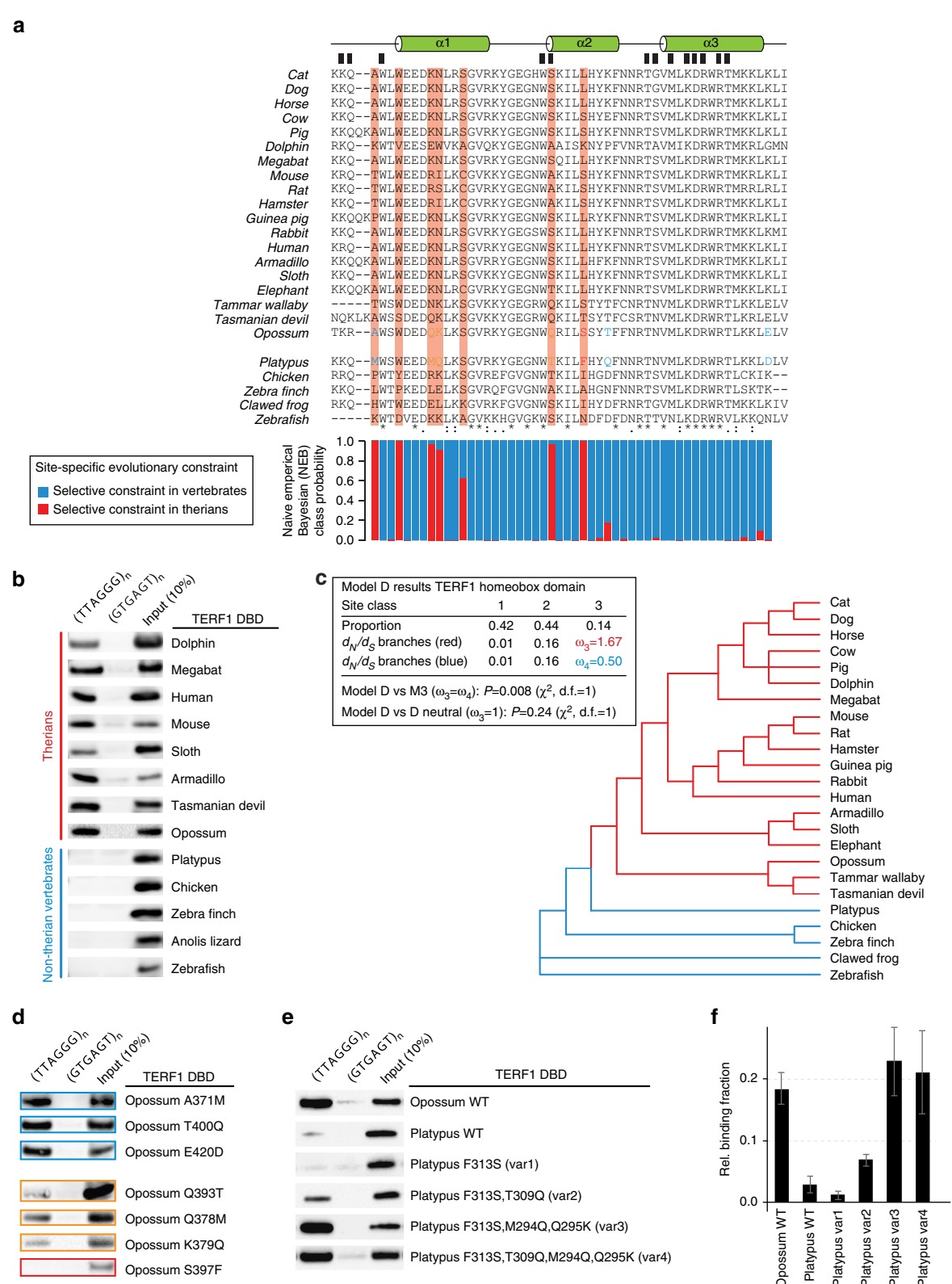

also highlight that the reasonable assumption to equate genetic and functional homology requires careful experimental testing and that proteins and protein complexes evolve dynamically in function and composition.

In conclusion, our approach combines the proliferation of sequenced genomes with the increasing power of interactomics screens to investigate functional and evolutionary important protein binding. Current synthesis and expression technologies allow for an easy production of baits, such as DNA sequences, RNA structures, (modified) peptides, protein domains or full-length proteins, while streamlined interaction screens can be performed in a large number of systems, given access to their proteomes. Thus phylointeractomics is capable to investigate the molecular evolution of domain-specific binding across any species and could serve as a blueprint for a future analysis of how full-length proteins evolve. It can provide experimental evidence for phylogenomics relationships and helps to extrapolate the results obtained in model organisms to a broader group of species.

## Methods

**Cell culture.** IMR90 (human), HeLa (human), NIH3T3 (mouse), PC12 (rat), Vx-2 (rabbit), LLC-PK1 (pig), MDCK (dog), CCL-141 (duck) and tasmanian devil cells were cultivated in $4.5\,g\,l^{-1}$ glucose, 4 mM glutamine, 1 mM sodium pyruvate Dulbecco's Modified Eagle's Medium (DMEM) supplemented with 10% fetal bovine serum (FBS; Sigma), $100\,U\,ml^{-1}$ penicillin and $100\,\mu g\,ml^{-1}$ streptomycin (Gibco), which is referred to as full DMEM hereafter. 104C1 cells (guinea pig) were cultured in RPMI-1640 medium supplemented with 10% FBS, 2 mM glutamine, $100\,U\,ml^{-1}$ penicillin and $100\,\mu g\,ml^{-1}$ streptomycin. Medium for the cultivation of OK (opossum) cells consisted of full DMEM complemented with 1% non-essential amino acids (PAA). Medium for culturing ZFTMA (zebra finch) cells was composed of full DMEM supplemented with 2% chicken serum. Chicken 6C2 cells were cultivated in DMEM with 10% FBS, $100\,U\,ml^{-1}$ penicillin and $100\,\mu g\,ml^{-1}$ streptomycin (Gibco), 1 mM sodium pyruvate, 1% non-essential amino acids and 0.4% chicken serum. Culture conditions of all above listed cell lines were 37 °C and 5% $CO_2$. Leibovitz's (L-15) medium (67%) supplemented with 10% FBS, $100\,U\,ml^{-1}$ penicillin and $100\,\mu g\,ml^{-1}$ streptomycin defined medium composition for the cultivation of speedy (frog) cells at 28 °C and 0% $CO_2$. BRF41 (zebrafish) cells were grown in L-15 medium, including 15% FBS, $100\,U\,ml^{-1}$ penicillin and $100\,\mu g\,ml^{-1}$ streptomycin at 33 °C and 0% $CO_2$. Medium for growing OLF-136 (medaka) cells was composed of DMEM supplemented with 15% FBS, $100\,U\,ml^{-1}$ penicillin and $100\,\mu g\,ml^{-1}$ streptomycin. Medaka cells were cultivated at 28 °C and 5% $CO_2$.

**Nuclear protein extraction.** Cells were harvested and nuclear extracts were prepared as previously described[37]. For extraction of nuclear extracts, cells were harvested and incubated in hypotonic buffer (10 mM Hepes, pH 7.9, 1.5 mM $MgCl_2$, 10 mM KCl) on ice for 10 min. Cells were transferred to a dounce homogenizer in hypotonic buffer supplemented with 0.1% Igepal CA630 (Sigma) and 0.5 mM DTT and lysed by 40 strokes. Nuclei were washed once in 1 × PBS and extracted in hypertonic buffer (420 mM NaCl, 20 mM Hepes, pH 7.9, 20% glycerol, 2 mM $MgCl_2$, 0.2 mM EDTA, 0.1% Igepal CA630 (Sigma), 0.5 mM DTT) for 2 h at 4 °C on a rotating wheel.

**Telomere pull-down.** Forward and reverse sequence oligonucleotides (25 µg) (see Supplementary Table 3) were diluted in annealing buffer (20 mM Tris-HCl, pH 7.5, 10 mM $MgCl_2$, 100 mM KCl), denatured at 95 °C and annealed by cooling. Annealed double-stranded oligonucleotides were incubated with 100 units T4 kinase (Life Technologies) for 2 h at 37 °C followed by incubation with 20 units T4 ligase overnight. Concatenated DNA strands were purified using phenol–chloroform extraction. Following biotinylation with desthiobiotin-dATP (Jena Bioscience) and 60 units DNA polymerase (Thermo), the biotinylated probes were purified using microspin G-50 columns (GE Healthcare). Telomeric or control DNA was immobilized on 500 µg paramagnetic streptavidin beads (Dynabeads MyOne C1, Life Technologies) on a rotation wheel for 30 min at room temperature. Subsequently, baits were incubated with 400 or 800 µg (frog) of nuclear extract in PBB buffer (150 mM NaCl, 50 mM Tris-HCl pH 7.5, 5 mM $MgCl_2$, 0.5% Igepal CA-630 (Sigma)) while rotating for 1.5 h at 4 °C. Sheared salmon sperm DNA (10 µg; Ambion) was added as a competitor for DNA binding. After three washes with PBB buffer, bound proteins were eluted in 1 × LDS sample buffer supplemented with 0.1 M DTT, boiled for 10 min at 70 °C and separated on a 10% NuPAGE Novex Bis-Tris precast gel (Life Technologies).

**MS data acquisition.** For in-gel digestion, samples were reduced in 10 mM DTT for 1 h at 56 °C followed by alkylation with 55 mM iodoacetamide (Sigma) for 45 min in the dark. Tryptic digest was performed in 50 mM ammonium bicarbonate buffer with 1 µg trypsin (Promega) at 37 °C overnight. Peptides were desalted on StageTips and analysed by nanoflow liquid chromatography on an EASY-nLC 1000 system (Thermo) coupled online to a Q Exactive Plus mass spectrometer (Thermo). Peptides were separated on a C18-reversed phase capillary (25 cm long, 75 µm inner diameter, packed in-house with ReproSil-Pur C18-AQ 1.9 µm resin (Dr Maisch) directly mounted on the electrospray ion source. We used a 90 min gradient from 2% to 60% acetonitrile in 0.5% formic acid at a flow of $200\,nl\,min^{-1}$. The Q Exactive Plus was operated with a Top10 MS/MS spectra acquisition method per MS full scan.

**MS data analysis.** The raw files were processed with MaxQuant[38] (version 1.4.0.8) against the ENSEMBL annotated protein and the genescan databases of the respective species (duck: BGI1.0.74; dog: CanFam3.1.71; guinea pig: cavPor3.75; zebrafish: Zv9.71; human: GRCh37.71; wallaby: Meug_1.0.74; opossum: BROADO5.71; mouse: GRCm38.71; rabbit: oryCun2.71; medaka: MEDAKA1.7; rat: Rnor_5.0.71; Tasmanian devil: DEVIL7.0.74; pig: Sscrofa10.2.71; zebra finch: taeGut3.2.4.71; xenopus: JGI_4.2.71; rabbit: oryCun 2.71) with the exception of axolotl where the Am2.0 database was used. Carbamidomethylation was set as fixed modification while methionine oxidation and protein N-acetylation were considered as variable modifications. The search was performed with an initial mass tolerance of 7 p.p.m. mass accuracy for the precursor ion and 20 p.p.m. for the MS/MS spectra in the HCD fragmentation mode. Search results were processed with MaxQuant and filtered with a false discovery rate of 0.01. The match between run option and the LFQ quantitation were activated.

**Bioinformatic analysis.** After peak detection and label-free quantitation was performed in MaxQuant, the files were further analysed using self-developed R and Python scripts. In detail, protein groups marked as reverse, contaminants or only identified by site were removed. A further filtering step removed protein groups that were not identified in at least three out of the four replicates at either bait. The missing values were imputed for each sample individually. The values were calculated with a beta distribution using the fitdistr function (MASS R package) fitted to 5% of the smallest values. Afterwards, a BLAST search was performed to map the human homologues to the different species. For creating the volcano plots, a two-tailed Welch's t-test to calculate the P value for each protein was used.

---

**Figure 3 | TERF1 acquired ability to bind telomeric DNA in the therian stem lineage.** (**a**) Sequence alignment of the TERF1 DBD of several species. Residues involved in DNA binding are marked by a black rectangle. Asterisks (*) indicate positions with fully conserved residues; colons (:) indicate exchanges with biochemical similar and periods (.) with related amino acid. Below each residue, a quantitative representation of the Naive Empirical Bayesian class probability derived from the branch-site modeling in **c** for selective constraints in therians (red; also in the background of these residues in the sequence alignment) and constraints in vertebrates (blue). Coloured residues refer to results of binding tests of amino-acid exchange variants from **d** with unchanged binding (blue), reduced binding (orange) or no binding (red). (**b**) Pull-down of the TERF1 DBD of various different vertebrate species with either telomeric repeats or a control oligonucleotide. (**c**) Phylogenetic relationship of the species used for substitution rate analysis. The colours denote the branch classifications, with red representing selective constraints across therians and blue representing selective constraints across vertebrates. Substitution rates were calculated using PAML[45] to obtain the non-synonymous to synonymous substitution rate ratio ($d_N/d_S = \omega$). $\omega$ values <1, =1 and >1 indicate purifying selection, neutral evolution and diversifying (positive) selection, respectively. A branch-site model (model D) was applied and compared with a homogeneous site model (discrete Model M3) and with a Model D that assumes neutral evolution for a predefined set of branches, representing our null hypothesis ($\omega = 1$). (**d**) Sequence-specific pull-down of single amino-acid exchange variants of the opossum TERF1 DBD. All seven residues exclusively found in platypus but not in therians were tested. (**e**) Sequence-specific pull-down of platypus TERF1 DBD variants using purified TERF1 DBDs. Combinations of the four identified residues from **d** were mutated to the opossum sequence to test whether their substitution can attribute TERF1 the capacity of directly binding to telomeric dsDNA. (**f**) Quantification of western blotting intensities from **e** ($n = 3$) shows similar binding affinity for platypus variants 3 and 4 compared with wild-type opossum TERF1-DBD. Mean with s.d. error bars.

Hit selection was based on $P$ values obtained from an analogous $t$-statistic with variance increased by a constant factor $S_0$ equal 0.6. The $P$ value cutoff indicated by dashed lines was set to 0.05. For creating the heat map, only proteins reported as enriched in at least five different species were considered. All plots where created using the ggplot2 and ggrepel R package. Data formatting and filtering was performed with the plyr and reshape2 R packages as well as base R commands.

**Telomeric repeat amplification protocol.** For all vertebrate species, 3.6 million cells were lysed in 100 μl lysis buffer (50 mM Tris-HCl (pH 8.0), 150 mM NaCl and 1% Igepal CA-630 (Sigma) supplemented with protease inhibitor). The quantitative telomeric repeat amplification protocol assay was carried out using GoTaq qPCR Master Mix (Promega) with both TS (5′-AATCCGTCGAGCAGAGTT-3′) and ACX primer (5′-GCGCGGCTTACCCTTACCCCTTACCCTAACC-3′) at 200 nM. Reactions were run on a ViiA 7 real-time PCR system (Thermo Fisher Scientific) with the following protocol: 25 °C for 20 min, 95 °C for 10 min and 40 cycles with 95 °C for 30 s, 60 °C for 30 s, and 72 °C for 1 min.

**Recombinant expression of TERF1 variants and binding test.** TERF1 DBDs were ordered as gene synthesis constructs (Genescript). The sequence was sub-cloned into the SLIC-compatible pETM44 vector via SLIC cloning[39] and expressed in *Escherichia coli* Rosetta at 25 °C. Amino-acid exchanges were introduced using site-directed mutagenesis[40] and validated by sequencing (GATC Biotech). Autoinduction was performed according to the published protocol[39] and bacteria were lysed in PBB buffer (150 mM NaCl, 50 mM Tris-HCl pH 7.5, 5 mM MgCl₂, 0.5% Igepal CA-630 (Sigma)) using a precooled Fastprep 24 system (Peqlab) with silica beads. Soluble supernatant of *E. coli* extracts with overexpressed recombinant proteins were used for telomere pull-downs. For purified protein domains, 100 ml cultures were autoinduced, harvested and treated with 5 mg lysozyme (Sigma) prior to sonication on ice with a Branson Sonifier 450 for 10 times 15 s with 1 min breaks in between. Protein purification was carried out as previously described[41]. In short, samples were centrifuged for 30 min at 3,500 $g$ and supernatant was filtrated with a 45 μm (Fisher Scientific) syringe filter. The supernatant was loaded on an equilibrated 1 ml HisTrap HP column (GE Healthcare). The column was washed with buffer containing 50 mM imidazole (Sigma) and the bound proteins were eluted in 250 μl fractions by buffer containing 500 mM imidazole (Sigma). The fractions were dialyzed overnight in storage buffer (0.5 M NaCl, 20 mM Tris-HCl pH 7.5, 5 mM MgCl₂, 10% glycerol and 1 mM DTT). The protein concentration was measured by Bradford assay (Bio-rad) and the purity of the elution fraction was assessed by polyacrylamide gel electrophoresis and Coomassie blue staining. Purified protein (10 μg) was incubated with 500 μg of paramagnetic streptavidin beads (Dynabeads, Thermo) coated with 600 nmol of biotinylated TTAGGG oligonucleotides for 2 h at 4 °C with slight agitation in PBB buffer (supplemented with 1 mM DTT and protease inhibitor). After three washes with PBB buffer, beads were transferred to a new Eppendorf tube and boiled in 1× LDS buffer (Thermo) containing 100 mM DTT for 10 min at 70 °C. Samples were loaded on a NuPage 4–12% Bis-Tris polyacrylamide gel (Thermo), which was run with 1× MES buffer (Thermo) for 45 min at 180 V.

**Western blotting.** Gels were blotted to nitrocellulose membranes (Protran83; Schleicher & Schuell) for 1 h constantly at 300 mA, blocked for 1 h at room temperature and incubated for 1 h with the anti-His₅ horseradish peroxidase-conjugated antibody following the manufacturer's instructions (Penta-His HRP Conjugate Kit, Qiagen). Membranes were washed twice in TBS-Tween-Triton buffer and once in TBS buffer for 10 min each. Detection was followed by incubating with enhanced chemiluminescence Prime Western Blotting Detection Reagent (GE-Healthcare). As a molecular weight standard, Seeblue 2 (Invitrogen) was used. For the purified domains, western blotting intensities were analysed using the integrated density function of ImageJ (https://imagej.nih.gov/ij/index.html).

**Multiple alignments and PAML statistical analysis.** DNA and protein sequences of human TERF1 and TERF2 orthologues from up to 24 vertebrate species were obtained from the ENSEMBL database[42] (release 75), including all species from the MS screen except axolotl for which there is currently no published genome available. To obtain multiple DNA sequence alignments, the corresponding protein sequences were aligned using MUSCLE[43] (version 3.8.31) and files were prepared with PAL2NAL[44] (version 14) to set up codon alignments and to remove gaps. Because whole-protein alignments for highly divergent species are difficult to obtain, we restricted the analyses to domain-specific alignments based on the human domain annotation. Here sequences were manually inspected and domains were separately analysed for the homeobox as well as TRFH domains of TERF1 and TERF2. Species for which the respective domain was not fully sequenced were excluded from further analysis. The exact species used for the analysis of the four different domains are depicted in the corresponding figure elements (Fig. 3c, Supplementary Fig. 4, Supplementary Table 1 and 2). Substitution rates were calculated using PAML[45] (version 4.7) to obtain the non-synonymous to synonymous substitution rate ratio ($d_N/d_S = \omega$). $\omega$ values $<1$, $=1$ and $>1$ indicate purifying selection, neutral evolution and diversifying (positive) selection, respectively. A branch-site model (model D) was applied and compared with a homogeneous site model (discrete model M3) and to a model D that assumes

neutral evolution for a predefined set of branches (for example, for the therian clade). In particular, we used a three-site class model, because we found a highly significant difference when compared with a discrete two-site class model, indicating heterogeneous levels of purifying selection within the protein domains. Significant differences between models were assessed by likelihood-ratio tests, which assume that the 2ΔlnL is approximately $\chi^2$ distributed with degrees of freedom being the number of free parameters.

**Code availability.** Scripts are available from the authors upon request.

**Data availability.** The mass spectrometry proteomics data have been deposited to the ProteomeXchange Consortium via the PRIDE[46] partner repository with the data set identifier PXD005517.

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

## Acknowledgements

We thank the following people for providing cell lines: Kai Simmons (dog), Janine Deakin (tammar wallaby and tasmanian devil), Nicolas Pollet (clawed frog), and Art Arnold (zebra finch), as well as Elly Tanaka for axolotl cells. Funding was provided by the Rhineland Pfalz Forschungsschwerpunkt GeneRED and the Deutsche Forschungsgemeinschaft (BU 2996/1) to F. Butter. D.K. is supported by the National Research Foundation Singapore and the Singapore Ministry of Education under its Research Centres of Excellence initiative. Work in the Buchholz laboratory was supported by the Excellence Initiative of the German Federal and State Governments (Institutional Strategy, measure 'support the best ZUK 64'). The research in the Mann laboratory received funding from the European Community's Health Seventh Framework Programme (FP7/2007–2013) under grant agreement no. 259867, the Center for Integrative Protein Science Munich and the Max Planck Society for the Advancement of Science. T.I.G. is supported by a Leverhulme Early Career Fellowship (Grant ECF-2015-453) and NERC grant N013832/1.

## Author contributions

D.K. and F. Butter initiated the research; D.K, M.S., A.B. and S.D. performed experiments; M.P.-R. and M.D. performed large-scale data analysis; T.I.G. analysed the evolutionary conservation of TERF domains; H.H. advised on vertebrate phylogeny and evolutionary analysis; F. Buchholz, M.M. and F. Butter supervised the research and contributed to the planning of the experiments; D.K., M.M. and F. Butter wrote the manuscript with input from all authors.

## Additional information

**Competing financial interests:** The authors declare no competing financial interests.

