## [Peer Review File · Nature Communications]

Reviewers' comments:

Reviewer #1 (Remarks to the Author):

These authors attempt to define the shelterin complexes in a large number of vertebrates using a telomeric DNA pull down assay combined with cutting edge mass spec analysis. The evolution of shelterin in vertebrates is of obvious interest. Their main result is that non-therian vertebrates lack the TRF1 component of the six-member shelterin complex. The absence of TIN2 in fish, chickens and frogs etc had already been noted but the finding that TRF1 might be lacking from telomeres in these animals would be significant. They then look at the TRF1 orthologs in the relevant species and show that E coli expressed DNA binding domains of these TRF1 orthologs failed to bind TTAGGG repeat DNA in vitro, apparently substantiating their claim. However, this reviewer is very concerned about this result since telomeric DNA binding has been shown for both chicken and frog TRF1 (in vivo and in vitro respectively). How well were these TRF1 DBDs expressed? Were they folded correctly? There is no data on the expression of the E coli recombinant proteins or a positive control for non-specific DNA binding (which should be OK for all the TRF1s). There are other issues of concern (below) but the claim that TRF1 is not a telomere binding protein outside therians is the main point.

Other points

The authors start out saying that the conservation of shelterin in vertebrates has been generally assumed but that no evidence for this is available. They give three references. One is on vertebrate evolution (nothing specific about shelterin), one is on transcriptional regulation by TRF2/Rap1 (not relevant) and one is on the evolution of CST (not relevant). In fact, my impression is that nobody in the field has made this assumption. Please remove this claim.

The authors claim the presence of a lot of other factors at telomeres. None of these are verified. Is this information actually of interest without further testing? Perhaps just focus this study on shelterin?

The whole study is based on the pull down assay. In this assay, a difference is measured between telomeric and scrambled baits. They add non-specific DNA during the binding to reduce background. But the 'non-specific' DNA used in excess in this assay is of a very odd. They use salmon sperm DNA at high concentration. Salmon sperm DNA has a very high TTAGGG content. It would be better to use E coli DNA. Proteins with a lower abundance/affinity for TTAGGGn but with high specificity might be competed out by the salmon sperm telomeric repeats.

RecQL1 is identified by their assay. They cite a paper to support a role for RecQL1 at telomeres, thus validating their approach. But in fact, RecQL1 has not been shown to play a role at 'regular' telomeres. The paper cited only examines telomeres in ALT cells where a very very large number of DNA repair proteins are at telomeres and affect telomeres that normally do not play a role in telomere biology. In contrast to RecQL1, other well established shelterin associated proteins are not detected (e.g. tankyrase 1, CST, BLM,

RTel1). So what is the relevance of the proteins that are, or are not, found by this method?

Reviewer #2 (Remarks to the Author):

General comments

This is a very interesting paper and seems to be quite novel, though the ideas have previously been theorized in general terms. However, some improvements to the manuscript can be made, as described below. We are somewhat concerned by some missing information in the methods description, for what is a relatively complex, multi-part work. If the following revisions are made (or otherwise addressed by the authors) and there are no unexpected results, we would recommend this manuscript for publication.

Major revisions

Some of the claims about the potential of phylointeractomics may be overstated, based on the evidence presented in the paper. The authors present a novel approach to evolution research, but their experiments largely constitute a pilot project limited in scope to protein-DNA interactions, using a highly conserved set of domains in two proteins that recognize a short, highly conserved, contiguous, exposed DNA consensus sequence. Performing a similar analysis using full-length proteins would be much more difficult, with additional complications for both the assay screening and bioinformatic analysis components. The authors should take care to acknowledge these complications in their discussion.

In Figure 3a, there are blue bars indicating that there is selective evolutionary constraint on every site amongst the vertebrates, but the experiments could not possibly have generated this conclusion, as there does not seem to be any invertebrate species included. We speculate that the authors have not used the correct statistical background to compare the selective constraint in therians against, by assuming that all positions across all species are each selected for at the same rate in vertebrates. This, however, is not a reasonable position: for example, column 7 of their alignment shows all Ws while column 8 shows a mixture of L and S, but both sites are supposedly equally constrained. Correcting this should only strengthen their conclusion, rather than weaken it.

Furthermore, not all 16 species are shown in alignment in Figure 3a, but the text that references the figure comment on an analysis across all 16 species, and the Bayesian analysis depicted in Figure 3a seems to have also been performed on all 16 species. This inconsistency should be corrected; otherwise, the figure is difficult to interpret.

For the bioinformatic analysis, the authors should not be using BLAST to identify homologs unless there is some specific reason they believe existing homolog databases (such as Ensembl or Inparanoid) are not suitable, and if so, they should specify why. In fact, the Ensembl database is used to retrieve TERF1 and TERF2 ortholog sequences for the multiple alignment.

For Figure 2a, proteins that do not exist in a particular species should be distinguished from proteins that do exist but were not detected with significant (positive) fold enrichment, as inconsistent enrichment is of key importance in evaluating the accuracy of the assay. As it stands, it is not possible to determine what a white box in the figure represents.

Why was negative enrichment not also included in Figure 2?

In the subsection titled "Multiple alignments and PAML statistical analysis," it is mentioned that "Species for which the respective domain was not fully sequenced were excluded from further analysis." The authors should make clear which species were excluded from analysis, and provide some interpretation on the significance of this exclusion.

Minor corrections

The authors should use the approved HGNC symbols TERF1 and TERF2, rather than TRF1 and TRF2.

BLAST should be capitalized as an acronym.

In Figure 2b, the white text is quite difficult to read when superimposed upon some of the lighter-coloured boxes.

Author point-by-point response

Reviewer: black

Authors: blue

Reviewer #1 (Remarks to the Author):

These authors attempt to define the shelterin complexes in a large number of vertebrates using a telomeric DNA pull down assay combined with cutting edge mass spec analysis. The evolution of shelterin in vertebrates is of obvious interest. Their main result is that non-therian vertebrates lack the TRF1 component of the six-member shelterin complex.

We appreciate that the reviewer considers our mass spectrometry approach “cutting edge” and our evolutionary study on shelterin of interest for telomere biology.

The absence of TIN2 in fish, chickens and frogs etc had already been noted but the finding that TRF1 might be lacking from telomeres in these animals would be significant.

To our knowledge there is no study showing the absence of TIN2 in fish, chicken and frog shelterin. We would be happy to include any literature evidence provided by the reviewer on this statement to improve our manuscript.

They then look at the TRF1 orthologs in the relevant species and show that E coli expressed DNA binding domains of these TRF1 orthologs failed to bind TTAGGG repeat DNA *in vitro*, apparently substantiating their claim. However, this reviewer is very concerned about this result since telomeric DNA binding has been shown for both chicken and frog TRF1 (*in vivo* and *in vitro* respectively).

We don't understand the concern, which seems to be rather a feeling than a data driven evaluation. We included an extended critical discussion on the existing data in the revised version of the manuscript based on the following information:

Xenopus TERF1 (xTERF1) has been functionally studied in one publication (Nishiyama et al., EMBO J, 2006). The authors see binding of *in vitro* transcribed/translated xTERF1 to chromatin or a plasmid containing TTAGGG repeats when added to mitotic extracts exclusively. Addition of *in vitro* transcribed/translated xTERF1 to interphase extracts (as used in our study) does not lead to chromatin/TTAGGG binding and even the mitotic phenotype is rather faint compared to xTERF2. While direct binding to TTAGGG was not assessed, the fact that a specific extract has to be present to achieve any enrichment suggests that the interaction is likely indirect. Clearly xTERF1 is not constitutively associated with telomeric DNA as it is in mammals, supporting our findings.

For chicken TERF1 (cTERF1) the literature is contradictory based on the only two available publications that functionally examined cTERF1. Okamoto et al., J Biol Chem, 2008 show that FLAG-cTERF1 does not localize to telomeres, but that they can localize a hybrid protein consisting of the chicken TRFH domain and the mouse TERF1 DBD (cmTERF1). In contrast to this, Cooley et al., Mol Biol Cell, 2009 show localization of myc-TERF1 in DT40 Terf1^{-/-} cells. While this issue needs further

investigation, neither of the two studies has actually tested direct TTAGGG binding of cTERF1. Furthermore, Cooley et al. report a lack of telomeric phenotypes in the DT40 Terf1^{-/-} cells and they noted that cTERF1 is not required for a functional telomere structure in contrast to mammalian TERF1, again in agreement with our findings.

How well were these TRF1 DBDs expressed? Were they folded correctly? There is no data on the expression of the E coli recombinant proteins or a positive control for non-specific DNA binding (which should be OK for all the TRF1s). There are other issues of concern (below) but the claim that TRF1 is not a telomere binding protein outside therians is the main point.

The absence of TERF1 binding to TTAGGG repeats has been noted in our original phylointeractomics screen which uses nuclear extracts from the corresponding species, i.e. the full-length, endogenous and correctly posttranslational modified version of each protein. The TERF1 DBDs validate this finding and highlight that the absence of TERF1 is due to a lack of direct TTAGGG binding.

All recombinant constructs expressed very well and only harbor a small His6-tag as seen by the Western input (Fig. 3b). While there always might be the consideration of misfolding in a single cloned TERF1 DNA-binding domain from one species, the clear number and phylogenetic association of the 8 binding (therians) versus 5 non-binding domains (non-therians) make this very unlikely.

Other points

The authors start out saying that the conservation of shelterin in vertebrates has been generally assumed but that no evidence for this is available. They give three references. One is on vertebrate evolution (nothing specific about shelterin), one is on transcriptional regulation by TRF2/Rap1 (not relevant) and one is on the evolution of CST (not relevant). In fact, my impression is that nobody in the field has made this assumption. Please remove this claim.

We only cited two papers for the shelterin evolution, the third on vertebrate evolution is used as a reference for an informed value of the evolutionary divergence between ray finned fish and mammals (“450 million years of divergent vertebrate evolution”), and thus of course has nothing specific about shelterin. We have now separated this citation in the sentence from the telomeric references to avoid any confusion.

Citations 7 (TERF2/Rap1) and 8 (evolution of CST) are clearly assuming that the shelterin complex is functionally conserved throughout the vertebrate lineage.

Ye et al. [7] write in their introductory paragraph that “the key protein component required for telomere capping is the shelterin complex which, **in vertebrates**, comprises six proteins: TRF1, TRF2, RAP1, TIN2, TPP1 and POT1.”

Likewise Price et al. [8] start their first paragraph with “**Vertebrate** telomeres are bound by six telomere-specific proteins that assemble into a complex termed shelterin.” Fig. 1a of their review displays the six shelterin proteins TERF1, TERF2, RAP1, TIN2, TPP1 and POT1 for “**Vertebrates**”.

Those are two very clear examples where established laboratories of the telomere community summarize this assumption within review articles in the last years.

Please note that we even consider this to be a reasonable assumption of phylogenomics.

The main point of our paper is that in many cases genetic homology might equate to functional homology but that this is not always the case as highlighted by the example of TERF1: the gene is present in all vertebrates but the direct binding affinity to TTAGGG is absent in non-therians. With phylointeractomics we pioneer a novel approach that can functionally assess such protein evolution. We also stress this point more clearly in the discussion of the revised manuscript.

The authors claim the presence of a lot of other factors at telomeres. None of these are verified.

The reviewer may have missed that besides the shelterin complex, already proven interactors include DCLRE1B, NR2C1/NR2C2, HOT1, RECQL1 and ZNF827, i.e. 6 additional proteins totaling 12 including the shelterin complex.

Is this information actually of interest without further testing? Perhaps just focus this study on shelterin?

The screen of 16 vertebrate species represents a very rich resource for the community and having consistently identified novel factors alongside various well-established telomere-binding proteins (see above) clearly suggests that these new candidates **may** be relevant for telomere biology. While this work is not focused on describing these new factors mechanistically (each one would likely justify a paper of its own), it is a high quality resource for other groups that should become available to the scientific community.

The whole study is based on the pull down assay. In this assay, a difference is measured between telomeric and scrambled baits. They add non-specific DNA during the binding to reduce background. But the 'non-specific' DNA used in excess in this assay is of a very odd. They use salmon sperm DNA at high concentration. Salmon sperm DNA has a very high TTAGGG content. It would be better to use E coli DNA. Proteins with a lower abundance/affinity for TTAGGGn but with high specificity might be competed out by the salmon sperm telomeric repeats.

This assessment is not quite correct. We want to point out that while the salmon genome as a vertebrate is capped by TTAGGG this makes up just the usual fraction of the genome. With 34 chromosomes in a haploid chromosome set, salmon sperm has 68 telomeres. While there is not much data available on salmon telomeres specifically, other fish species have reported telomere length with an average of ~10kb (e.g. see Henriques et al., PLoS Genetics, 2013). Out of a genome with ~3 billion nucleotides telomeric DNA thus comprise <0.001% of the salmon genomic DNA. Even with conservative estimates, this creates a situation in which our bait TTAGGG is at least 10,000 times in excess of the TTAGGG content of the non-specific competitor. Thus, no TTAGGG-specific factor is prevented from binding to our bait sequence.

Using E. coli DNA will have no effect on the results, while it has other theoretical limitations. The E. coli genome is by far less complex than any higher eukaryotic genome and has another GC content.

RecQL1 is identified by their assay. They cite a paper to support a role for RecQL1 at telomeres, thus validating their approach. But in fact, RecQL1 has not been shown to

play a role at 'regular' telomeres. The paper cited only examines telomeres in ALT cells where a very very large number of DNA repair proteins are at telomeres and affect telomeres that normally do not play a role in telomere biology. In contrast to RecQL1, other well established shelterin associated proteins are not detected (e.g. tankyrase 1, CST, BLM, RTEL1). So what is the relevance of the proteins that are, or are not, found by this method?

RECQL1 is only one example of various factors that we cite as previously associated with telomeres (together with TERF1, TERF2, RAP1, TIN2, POT1, TPP1, NR2C1, NR2C2, DCLRE1B, HOTT1 & ZNF827). Nevertheless, while RECQL1 has indeed primarily been studied in the ALT cell line U2OS in a fairly recent publication (Popuri et al., NAR, 2014) the authors also show that RECQL1 associates with a fraction of telomeres in HeLa cells and that “similar to U2OS cells, RECQL1 depletion caused telomere shortening and an accumulation of DNA DSB foci at TERF1 sites” in these cells. The authors conclude that “these results indicate that RECQL1 may bind to telomeres in telomerase-positive cells, thereby regulating telomere integrity”.

Concerning the mentioned proteins (tankyrase1, CST, BLM, RTEL1), we would like to highlight that we use a DNA-binding pull-down that primarily identifies direct binding proteins or tight interaction partners. We do not state anywhere that we want to identify every protein that can indirectly and/or transiently interact with telomeric DNA. Such an expectation would be presumptuous as we use a biochemical purification method.

Reviewer #2 (Remarks to the Author):

General comments

This is a very interesting paper and seems to be quite novel, though the ideas have previously been theorized in general terms. However, some improvements to the manuscript can be made, as described below. We are somewhat concerned by some missing information in the methods description, for what is a relatively complex, multi-part work. If the following revisions are made (or otherwise addressed by the authors) and there are no unexpected results, we would recommend this manuscript for publication.

We thank the reviewer for the positive feedback. Additionally, to the detailed points below, we have extended and clarified the method section.

Major revisions

Some of the claims about the potential of phylointeractomics may be overstated, based on the evidence presented in the paper. The authors present a novel approach to evolution research, but their experiments largely constitute a pilot project limited in scope to protein-DNA interactions, using a highly conserved set of domains in two proteins that recognize a short, highly conserved, contiguous, exposed DNA consensus sequence. Performing a similar analysis using full-length proteins would be much more difficult, with additional complications for both the assay screening and bioinformatic

analysis components. The authors should take care to acknowledge these complications in their discussion.

Yes, we here show a pilot experiment investigating the telosome in 16 different species as a pilot. However, there is an unfortunate misunderstanding of our technique. In phylointeractomics we use cellular material from these diverse species. In this extract, the full-length proteins are expressed at endogenous level, including all spliced isoforms and with the correct posttranslational modifications (Figure 1c).

We use the TERF1 DNA-binding domains (Figure 3) to validate our phylointeractomics hypothesis and to further narrow the phylogenetic branch-point as we could not get enough biomass from platypus cells. Thus, we have in fact used full-length proteins in the screen and will also do this in future phylointeractomics experiments. As we identify proteins by mass spectrometry, the availability of full-length proteins in the extract allows very accurate identification using the protein database of each species. Our current major limitation is to get access to species with a gene-model, but given the enormous progress in sequencing of all branches of life, this restriction will resolve soon. We have already initiated projects to study the evolution of polyA-binding proteins (RNA-protein interactions), histone modification binding proteins (peptide-protein) and conserved promoter/enhancer elements (DNA-protein interactins). We have done this previously comparing two species (Butter et al., 2009; Viturawong et al., 2013; Casas-Vila et al., 2015; Bluhm et al., 2016), but tracing binding events to the evolutionary origin is much more beneficial to truly understand evolution of protein functionality as demonstrated in this pilot experiment.

In Figure 3a, there are blue bars indicating that there is selective evolutionary constraint on every site amongst the vertebrates, but the experiments could not possibly have generated this conclusion, as there does not seem to be any invertebrate species included. We speculate that the authors have not used the correct statistical background to compare the selective constraint in therians against, by assuming that all positions across all species are each selected for at the same rate in vertebrates. This, however, is not a reasonable position: for example, column 7 of their alignment shows all Ws while column 8 shows a mixture of L and S, but both sites are supposedly equally constrained. Correcting this should only strengthen their conclusion, rather than weaken it.

The reviewer is correct; no invertebrates were included in the bioinformatics analysis. The existence of TERF genes in invertebrates seems limited. For instance, ENSEMBL only reports TERF homologs in *Ciona* sea squirts and reciprocal BLASTing is not yielding many homologs either. What is more, invertebrates seem to have only a single TERF gene. As our analysis is specific to either of the two paralogs, it is thus not possible to integrate TERF into our analysis to detect selective constraints within the vertebrate lineage using site-specific substitution rates.

Using the blue bars, we show that evolutionary constraint applies for therians and non-therian vertebrates (site classes 0 and 1 in the box in Figure 3c). In the mentioned example site column 7 has a higher probability than column 8 to be rather site class 0 ($d_N/d_S=0$) than site class 1. For visibility reasons we have merged the two site classes as we are interested in site class 2.

Furthermore, not all 16 species are show in alignment in Figure 3a, but the text that

references the figure comment on an analysis across all 16 species, and the Bayesian analysis depicted in Figure 3a seems to have also been performed on all 16 species. This inconsistency should be corrected; otherwise, the figure is difficult to interpret.

We thank the reviewer for pointing this out and we have corrected this inconsistency. The substitution rate analysis for the TERF1 DBD was in fact performed on 24 vertebrate species. We have now included the alignment for all these species in Fig. 3a.

For the bioinformatic analysis, the authors should not be using BLAST to identify homologs unless there is some specific reason they believe existing homolog databases (such as Ensembl or Inparanoid) are not suitable, and if so, they should specify why. In fact, the Ensembl database is used to retrieve TERF1 and TERF2 ortholog sequences for the multiple alignment.

We used the ENSEMBL database for establishing homology. However, the database is not complete. Only for missing ENSEMBL homologs (now written in grey font), we double-checked using reciprocal BLAST searches and found putative 1-1 homologies in some cases (e.g. for TIN2 in clawed frog, medaka and zebrafish but not in duck and zebra finch). We want to ensure not to erroneously report absence of homologs based on incomplete databases and therefore included these putative 1-1 homologs into Fig. 2b. We have also adapted our figure legend to make this clearer and would like to thank the reviewer for spotting this ambiguity.

For Figure 2a, proteins that do not exist in a particular species should be distinguished from proteins that do exist but were not detected with significant (positive) fold enrichment, as inconsistent enrichment is of key importance in evaluating the accuracy of the assay. As it stands, it is not possible to determine what a white box in the figure represents.

We agree with the reviewer and changed Fig. 2a accordingly. Every protein that was detected in our proteomics screen but not significantly enriched on telomeric DNA is now depicted with a blue square.

Why was negative enrichment not also included in Figure 2?

We evaluated whether to include quantitative values for negative enrichment beyond the blue squares, but realized that this would lead to misleading data. For instance, TERF1 in axolotl has a fold enrichment of ~ 3.6 . However, this value is not statistically significant and thus has to be interpreted as not specifically enriched. If we were plotting all detected proteins without consideration for the p-value the heatmap would visually suggest false positives. Therefore, we decided to only have a yes-no answer with blue vs. white squares to distinguish between proteins that were detected but not specifically bound to telomeric DNA and proteins that were not detected at all. Nevertheless, the exact fold enrichments and p-values for the candidates as well as all proteins detected are provided as Suppl. Tables S1 & S2.

We would also like to highlight that we chose our control sequence as a repetitive sequence that has the same base content as TTAGGG but in a different order. Negative enrichment thus means a higher affinity to GTGAGT, but there is no direct biological readout for this. Therefore, to avoid any confusion in Fig. 1c and to make the color-

coding more consistent between Fig. 1c and 2a we removed the dashed cut-off line for the control sequence and colored the factors specifically binding to GTGAGT in blue.

In the subsection titled "Multiple alignments and PAML statistical analysis," it is mentioned that "Species for which the respective domain was not fully sequenced were excluded from further analysis." The authors should make clear which species were excluded from analysis, and provide some interpretation on the significance of this exclusion.

We would like to thank the reviewer for pointing out this inconsistency in our documentation. For this analysis we aimed to include all 15 species from our mass spectrometry screen with a published genome (for Axolotl we had to resort to an EST database) and we wanted to include as many additional vertebrates as possible since we were not limited by the availability of cell lines for this *in silico* analysis. However, gene annotations are not complete in all available genomes. Therefore, analysis for the four different domains included a different number of species, namely:

TERF1 DBD: 24 species

TERF1 TRFH domain: 22 species

TERF2 DBD: 24 species

TERF2 TRFH domain: 18 species

For TERF1 the phylogenetic trees represent the species that were included in the respective analysis (Fig. 3c & Fig. S1) and for TERF2 we list the species with Suppl. Tables 3 & 4. As the exclusion of particular species is due to technical reasons, the general conclusions of the substitution rate analysis should not be affected.

Minor corrections

The authors should use the approved HGNC symbols TERF1 and TERF2, rather than TRF1 and TRF2.

We changed to the HGNC symbols TERF1 and TERF2 as suggested.

BLAST should be capitalized as an acronym.

BLAST is now capitalized at all occurrences.

In Figure 2b, the white text is quite difficult to read when superimposed upon some of the lighter-coloured boxes.

We modified the figure for better reading and now depict these values in grey (please see above for the explanation on why these values are not written in black).

Reviewers' comments:

Reviewer #2 (Remarks to the Author):

The authors have addressed most of the concerns from our previous review.

There are minor concerns with two responses:

Regarding the response about overstated claims: We do not disagree with the substance of this response, but we believe it misses the point made in the initial review. Notably, the method demonstrated in this paper only analyzes the evolutionary relationship between the DNA-binding affinity and sequence of a single domain on a full-length protein, and not the overall biomolecule-binding affinity and sequence co-evolution of the entire protein, or even other domains within the protein. The identification of molecular binding in this broader context, especially to other proteins, remains a substantial hurdle.

We point to, for example, the concluding paragraph in the main text, which references "full-length proteins" and "streamlined interaction screens," and then goes on to say that "phylointeractomics is capable to investigate the molecular evolution of protein binding across any species." We do not believe that this paper directly demonstrates this possibility, though it is a useful blueprint for how the analysis could be conducted, if the experimental data could be produced. This could be addressed briefly in the discussion.

Regarding the response about Figure 3a:

We originally did not understand the bottom half of Figure 3a to mean what was intended. The response has clarified for us that that graph was meant to depict the relative strength of the two selective constraint hypotheses, across vertebrates or across therians. The figure itself would seem to suggest that the null hypothesis was no selective constraint at all. We would suggest a change to the legend/axis label, and a revision of the figure caption to clarify what the graph depicts.

Reviewer #3 (Remarks to the Author):

Review: Phylointeractomics reconstructs functional evolution of protein binding. (Kappei et al.)

Summary: This study seeks to determine the extent of evolutionary conservation of the vertebrate telosome. An unbiased interactomic screen was performed to find which proteins are enriched on immobilized repetitive vertebrate double-stranded telomere TTAGGG DNA (compared to control GTGAGT) when incubated with 16 different vertebrate nuclear extracts. Bioinformatic analyses of recovered peptides suggested that shelterin-telomere binding was largely conserved in many vertebrates, but its components and organization was extremely different in other vertebrates. In particular the authors suggest that the TRF1 telomere binding is not conserved in some organisms, such as chicken, zebrafish and platypus. However, when residues are mutated in the TRF1 binding domain of organisms

that don't bind the telomere (e.g. Platypus) to TRF1 residues that do (e.g. Opossum) some telomeric binding is restored in vitro. The authors also report the identification of several proteins that were not previously known to associate with telomeres.

This is an ambitious and interesting study that combines biochemistry, mass spec and bioinformatics to understand the evolution of the telosome and identify new telomeric proteins. The major issues itemized below should be addressed before it is published, otherwise it should be rejected with the possibility of a resubmission that concentrated on analysis of the novel telomere interactors.

Major points:

1) The main limitation of this study is that it is dependent on the assumption that telosome interactions form on the biotinylated telomeric DNA in vitro as they do in vivo. Telomere protein binding kinetics, stoichiometry, and organization in the cell are complex, dynamic and regulated. It is not necessarily true that the proteins that the authors are recovering under in vitro conditions accurately reflect in vivo telomere protein associations and arrangements. Therefore, this IP/Mass Spec technique seems best suited to identifying novel protein-telomere interactions (which the authors report). The potential new telomere-interacting proteins should then be characterized further by other experimental means to determine their function.

Because it is impossible to know if facets of in vivo telosome-telomere interaction are preserved in the authors' in vitro system it is difficult to make conclusions from interactions that do not occur. For example, the authors conclude that TRF1-telomere binding evolved in placental and marsupial vertebrates, and, therefore, that avian TRF1 does not associate with telomeres (Fig. 3b). However, it has been previously reported that chicken TRF1 does bind telomeres (PMID:18587156).

2) The authors perform their pull down assay with a mixture of vertebrate cell lines that have telomerase activity and do not have telomerase activity. They state that this is useful information as some factors may associate when telomerase is active, but not in cells without telomerase activity (i.e. human HOTA1). This is a good reason, but it is impossible to make comparisons when only one type of cell line is used for each species. This is especially problematic given the concern listed in major point 1, i.e. it is difficult to know if the absence of a protein is genuinely due to the biology of the cell line or the in vitro binding system. At the very least, the experiment should have been performed with HeLa (telomerase +; PMID:23685356) to compare to the IMR90 line, as well as telomerase + and - cell lines for the other vertebrate species, if available. It would be particularly useful to have a telomerase + and - line for a representative of the TRF1-telomere binding group (e.g. humans or mouse) and non-TRF1-telomere binding group (e.g. chicken or zebrafish).

3) The TRF1-DNA binding experiments described in Fig. 3d and 3e should include wild-type protein controls, quantitation and equal amounts of protein for each mutant. The experiments should be performed in triplicate with statistical analysis and demonstrations of each protein's purity shown. Also, a description of how the binding assays were performed (conditions, quantities, etc) is required for the Methods section.

Minor points:

- 1) On page 4, line 13. The authors state that only TRF1, TRF2, HOTA1 and POT1 are known to directly bind telomeric DNA. The authors should note that the CST complex also specifically binds telomeric DNA (PMID: 22763445).
- 2) Because the authors' work is dependent on telosome interactions forming in vitro, they may want to compare their human telomere repeat in vitro interactors with the large number of shelterin interacting proteins identified in vivo by BiFC (PMID: 21044950).
- 3) In many places it is not clear which species the authors are discussing. It would be helpful to identify the protein species (e.g. hHOTA1 for human HOTA1). Alternatively, if they are referring to the same protein from more than one species to specifically state it.
- 4) A column to the side of Fig. 2a denoting telomerase + or - activity for each cell line would be useful to the reader.

Comments on Reviewer #1:

- I agree with reviewer #1 that it is a problem that TRF1 has been shown to bind telomeres in non-mammalian vertebrates (contradicting the authors' finding). I don't think the authors' rebuttal that the reviewer's concerns are a "feeling" is justified. The authors state that two publications have examined chicken TRF1 telomere binding and that they are contradictory. They acknowledge that Cooley et al. showed myc-TRF1 telomere localization in chicken DT40 cells. However, they state that Okamoto et al. "show that FLAG-cTERF1 does not localize to telomeres." However, this is not true, Okamoto et al. state that it does and report it as data not shown:

"On the other hand, we obtained evidence that a chicken TRF1 homologue (cTRF1) localized to telomeres in a chicken B cell line, DT40 and that a GFP-tagged mTIN2 did not form telomeric foci in the same cells (data not shown)." (PMID: 18587156).

- I agree with reviewer #1 that there are questions about the in vitro TRF1 DBD binding assays (major point 3 above). I think the experiments need to be performed again with more rigor.

- I think that the author's responses to the other points made by reviewer 1 are satisfactory.

Author point-by-point response

Reviewer: black

Authors: blue

Reviewers' comments:

Reviewer #2 (Remarks to the Author):

The authors have addressed most of the concerns from our previous review.

There are minor concerns with two responses:

Regarding the response about overstated claims: We do not disagree with the substance of this response, but we believe it misses the point made in the initial review. Notably, the method demonstrated in this paper only analyzes the evolutionary relationship between the DNA-binding affinity and sequence of a single domain on a full-length protein, and not the overall biomolecule-binding affinity and sequence co-evolution of the entire protein, or even other domains within the protein. The identification of molecular binding in this broader context, especially to other proteins, remains a substantial hurdle.

We point to, for example, the concluding paragraph in the main text, which references "full-length proteins" and "streamlined interaction screens," and then goes on to say that "phylointeractomics is capable to investigate the molecular evolution of protein binding across any species." We do not believe that this paper directly demonstrates this possibility, though it is a useful blueprint for how the analysis could be conducted, if the experimental data could be produced. This could be addressed briefly in the discussion.

We would like to thank the reviewer for this clarification as we misunderstood the point made in the initial review: While our extracts contain full-length proteins and hereby interrogate entire proteomes for binding to telomeric DNA, we indeed only test for the affinity of the DNA-binding domain and hence evolutionary relationship between the TTAGGG motif and single domains within the full-length proteins. We have now amended our concluding paragraph accordingly: "Thus, phylointeractomics is capable to investigate the molecular evolution of domain-specific binding across any species and could serve as a blueprint for a future analysis of how full-length proteins evolve."

Regarding the response about Figure 3a:

We originally did not understand the bottom half of Figure 3a to mean what was intended. The response has clarified for us that that graph was meant to depict the relative strength of the two selective constraint hypotheses, across vertebrates or across therians. The figure itself would seem to suggest that the null hypothesis was no selective constraint at all. We would suggest a change to the legend/axis label, and a revision of the figure caption to clarify what the graph depicts.

This is exactly what Panels 3a/c are meant to represent. We have adapted the labeling as well as the figure caption to avoid remaining ambiguities.

Reviewer #3 (Remarks to the Author):

Review: Phylointeractomics reconstructs functional evolution of protein binding. (Kappei et al.)

Summary: This study seeks to determine the extent of evolutionary conservation of the vertebrate telosome. An unbiased interactomic screen was performed to find which proteins are enriched on immobilized repetitive vertebrate double-stranded telomere TTAGGG DNA (compared to control GTGAGT) when incubated with 16 different vertebrate nuclear extracts. Bioinformatic analyses of recovered peptides suggested that shelterin-telomere binding was largely conserved in many vertebrates, but its components and organization was extremely different in other vertebrates. In particular the authors suggest that the TRF1 telomere binding is not conserved in some organisms, such as chicken, zebrafish and platypus. However, when residues are mutated in the TRF1 binding domain of organisms that don't bind the telomere (e.g. Platypus) to TRF1 residues that do (e.g. Opossum) some telomeric binding is restored *in vitro*. The authors also report the identification of several proteins that were not previously known to associate with telomeres.

This is an ambitious and interesting study that combines biochemistry, mass spec and bioinformatics to understand the evolution of the telosome and identify new telomeric proteins. The major issues itemized below should be addressed before it is published, otherwise it should be rejected with the possibility of a resubmission that concentrated on analysis of the novel telomere interactors.

Major points:

1) The main limitation of this study is that it is dependent on the assumption that telosome interactions form on the biotinylated telomeric DNA *in vitro* as they do *in vivo*. Telomere protein binding kinetics, stoichiometry, and organization in the cell are complex, dynamic and regulated. It is not necessarily true that the proteins that the authors are recovering under *in vitro* conditions accurately reflect *in vivo* telomere protein associations and arrangements. Therefore, this IP/Mass Spec technique seems best suited to identifying novel protein-telomere interactions (which the authors report). The potential new telomere-interacting proteins should then be characterized further by other experimental means to determine their function.

As the reviewer rightfully points out our approach is based on *in vitro* reconstitution and therefore does not aim to recapitulate the entire *in vivo* telomere composition. Rather it is a screening method that is very potent at identifying direct sequence-specific DNA-binding proteins and their strong interaction partners. This is clearly shown e.g. by the identification of the entire shelterin complex in the therian lineage. While we are providing a valuable resource of novel putative telomere-interacting proteins to the community, it is beyond the scope of this manuscript to determine the function of these novel factors. The major aim of this study is not the biological function of individual proteins at telomeres, but to (1) establish a systematic and quantitative approach to functionally study protein evolution, (2) follow the protein evolution in molecular detail in a proof-of-concept study and (3) to illustrate that one cannot simply equate genetic and functional homology, but that rather an experimental systematic approach is required.

Because it is impossible to know if facets of *in vivo* telosome-telomere interaction are preserved in the authors' *in vitro* system it is difficult to make conclusions from interactions that do not occur. For example, the authors conclude that TRF1-telomere binding evolved in placental and marsupial vertebrates, and, therefore, that avian TRF1 does not associate with telomeres (Fig. 3b). However, it has been previously reported that chicken TRF1 does bind telomeres (PMID:18587156).

Again, we agree that we are not assaying the entire *in vivo* composition at telomeres. However, TERF1 is not only a direct telomere-binding protein but a *bona fide* telomere marker that is constitutively bound to telomeres. Therefore, it is a big change in evolutionary terms if such a protein loses the intrinsic ability to directly bind to this DNA element.

As the cited publication (PMID:18587156) indicates that TERF1 can localize to telomeres *in vivo*, we carried out our telomere pull-down using nuclear protein extracts from chicken cells to provide experimental data to the discussion (new **Suppl. Fig. 1b**). Indeed, TERF1 was enriched on telomeric DNA in this experiment. This is in agreement with the reported literature. However, our direct binding assay based on the recombinantly expressed cTERF1 DNA-binding domain indicates that also in chicken TERF1 cannot directly bind to telomeric DNA. These data suggest that in chicken TERF1 might be recruited to telomeres via protein-protein interactions. Furthermore, the chicken example does not represent the entire avian lineage since with zebra finch extracts TERF1 was identified as a background binder in our proteomics screen. Considering further that in xenopus TERF1 can only associate with telomeres in the presence of mitotic protein extract (PMID:16424898), the association of TERF1 in non-therian vertebrates seems heterogeneous and it will be very interesting to understand the exact underlying biology in the future.

We now included the new data on chicken TERF1 and also highlight the above described aspect more clearly to distinguish unambiguously between “associated with telomeres” and “binding to telomeres”.

2) The authors perform their pull down assay with a mixture of vertebrate cell lines that have telomerase activity and do not have telomerase activity. They state that this is useful information as some factors may associate when telomerase is active, but not in cells without telomerase activity (i.e. human HOTA1). This is a good reason, but it is impossible to make comparisons when only one type of cell line is used for each species. This is especially problematic given the concern listed in major point 1, i.e. it is difficult to know if the absence of a protein is genuinely due to the biology of the cell line or the *in vitro* binding system. At the very least, the experiment should have been performed with HeLa (telomerase +; PMID:23685356) to compare to the IMR90 line, as well as telomerase + and - cell lines for the other vertebrate species, if available. It would be particularly useful to have a telomerase + and - line for a representative of the TRF1-telomere binding group (e.g. humans or mouse) and non-TRF1-telomere binding group (e.g. chicken or zebrafish).

We agree with the reviewer that ideally we would have tissue matched cell lines and putatively even isogenic cell line pairs that could address specific biological aspects (e.g. expression of telomerase, telomere length, presence of ALT mechanism etc.). At the same time we would like to point out that the current dataset represents >1000 hours of uninterrupted proteomics measurements and that the number of available cell

lines throughout the phylogenetic tree with good protein annotated species is currently extremely limited.

We had only mentioned HOTA1 as an example in which putative cell context specific effects correlates with the previously described function of HOTA1 as a differential telomere-binding protein that associates with telomeres in particular in settings with active telomere elongation (PMID: 23685356). To test whether this holds true in our setting we now also performed DNA pull-downs with HeLa cells as suggested by the reviewer (new **Suppl. Fig. 1a**). Indeed, we identify HOTA1 as a specific telomere-binder when using HeLa extracts, which is not the case for the IMR90 extracts. In addition, we determined the presence of telomerase activity by TRAP for the new cell lines used in this study (**Fig. 2a**, new **Suppl. Fig. 2**).

3) The TRF1-DNA binding experiments described in Fig. 3d and 3e should include wild-type protein controls, quantitation and equal amounts of protein for each mutant. The experiments should be performed in triplicate with statistical analysis and demonstrations of each protein's purity shown. Also, a description of how the binding assays were performed (conditions, quantities, etc) is required for the Methods section.

The data on the opossum mutations (**Fig. 3d**) is solely meant as a screen to identify which exchanges would be most likely to achieve our aimed gain of function in the platypus TRF1-DNA binding domain. However, we agree with the reviewer completely that for investigation of the gain of function mutation which is a strong claim, the effort for quantitative data should be made. Thus, we purified the platypus variant domains and included the requested controls (platypus wt and opossum wt). We perform statistical analysis (mean and standard deviation) on triplicate experiments (new **Fig. 3e**), show a representative blot (new **Fig. 3f**) and the requested purity by Coomassie staining (new **Suppl. Fig. 4**). These new experiments are now described with the requested information in a new section of Material and Methods.

Minor points:

1) On page 4, line 13. The authors state that only TRF1, TRF2, HOTA1 and POT1 are known to directly bind telomeric DNA. The authors should note that the CST complex also specifically binds telomeric DNA (PMID: 22763445).

We now cite this manuscript accordingly.

2) Because the authors' work is dependent on telosome interactions forming in vitro, they may want to compare their human telomere repeat in vitro interactors with the large number of shelterin interacting proteins identified in vivo by BiFC (PMID: S21044950).

We compared our telomere-binding candidates with the mentioned BiFC data as well as with various other studies for additional telomeric proteins in human cells, namely PICH, Q-TIP, and several cross-linking and non-cross-linking shelterin IPs that are all listed in the TeloPIN database (PMID: 25792605). We generated Venn diagrams (new **Suppl. Fig. 3**) for the overlaps with our study and between the different studies and discuss this in the manuscript. In brief, our set of candidates is smaller as we very specifically screen for direct TTAGGG binders and their tight interaction partners, but

in general the clearest overlap is seen with the PICCh approach. It is noteworthy that this is the only other approach that uses DNA as the bait while all other studies are shelterin-centered. Among these different studies the overlap is generally low, with usually only the shelterin proteins found in common when comparing 3 studies.

3) In many places it is not clear which species the authors are discussing. It would be helpful to identify the protein species (e.g. hHOT1 for human HOT1). Alternatively, if they are referring to the same protein from more than one species to specifically state it.

We highlighted the species context more clearly throughout the manuscript.

4) A column to the side of Fig. 2a denoting telomerase + or - activity for each cell line would be useful to the reader.

This has been added according to the major comment 2 above.

Comments on Reviewer #1:

- I agree with reviewer #1 that it is a problem that TRF1 has been shown to bind telomeres in non-mammalian vertebrates (contradicting the authors' finding). I don't think the authors' rebuttal that the reviewer's concerns are a "feeling" is justified. The authors state that two publications have examined chicken TRF1 telomere binding and that they are contradictory. They acknowledge that Cooley et al. showed myc-TRF1 telomere localization in chicken DT40 cells. However, they state that Okamoto et al. "show that FLAG-cTERF1 does not localize to telomeres." However, this is not true, Okamoto et al. state that it does and report it as data not shown: "On the other hand, we obtained evidence that a chicken TRF1 homologue (cTRF1) localized to telomeres in a chicken B cell line, DT40 and that a GFP-tagged mTIN2 did not form telomeric foci in the same cells (data not shown)." (PMID: 18587156).

We have already outlined our major extension to the chicken TERF1 above. Nevertheless, we would like to thank the reviewer for helping us to clarify this point. While we regret that there is no primary data available for cTERF1 localization in chicken cells, the conclusion is that cTERF1 can putatively localize to telomeres in chicken cells but not in mouse cells, which requires the cmTERF1 fusion. This is in fact compatible with an indirect recruitment of cTERF1 to telomeres. In agreement with this, we also enrich cTERF1 in our pull-down assay from complex nuclear protein extracts, but recombinantly expressed cTERF1-DBD does not directly bind to telomeric DNA.

This underscores that TERF1 is only a bona fide direct TTAGGG binder that is found at all telomeres constitutively in the therian lineage.

- I agree with reviewer #1 that there are questions about the in vitro TRF1 DBD binding assays (major point 3 above). I think the experiments need to be performed again with more rigor.

Please see major point 3 above.

- I think that the author's responses to the other points made by reviewer 1 are

satisfactory.

We thank the reviewer for stepping in for reviewer 1 and hope that we have also addressed all her/his comments adequately.

Reviewers' comments:

Reviewer #3 (Remarks to the Author):

Phylointeractomics reconstructs functional evolution of protein binding

Kappie, et al.

The authors have addressed most of the concerns raised in my original review. The in vitro binding data with the chicken TERF1 BD is a good addition. The new data makes a stronger case for the interesting hypothesis about the evolution of shelterin-telomere interaction. However, the main criticism is still relevant. The authors claim that the complexes that they are building on telomeric sequence in vitro necessarily reflect the in vivo arrangements. The abstract states, "To investigate such functional evolution, we here combine phylogenomics with interaction genomics." I cannot find a description of functional genomics, which includes analysis of protein interactions that are built in vitro from a lysate on a substrate. For example, the review article about functional genomics (PMID:17593931) that is cited by the authors only describes functional genomics as interactions that are observed in complexes that are formed in the cell. This is not a minor distinction. Immuno-precipitation of complexes combined with MS is a powerful method that determines relationships between proteins in the cell. Adding a lysate to build complexes in vitro may lose many of the in vivo relationships and interactions that do not usually form due to temporal or spatial differences. Further it could amplify weak interactions or potentially create interactions that do not naturally occur in vivo. PMID:17593931 states post translational modifications determine complex formation. The cleared extract generated using the Dignam et al., 1983 nuclear extract protocol could concentrate free telomere proteins that are modified to not interact with the telomere as the chromosomes are spun out. Therefore, it is possible that the complexes built and interactions formed in vitro do not reflect the interactions that occur in the cell and are not "unbiased", as claimed in the manuscript. Furthermore, the lack of G-tail in the substrates used to build shelterin from the different species could bias against organisms that may be dependent on POT1 binding for the association of the other shelterin and telomeric proteins. Finally, the Dignam et al protocol was developed for human cells and may not be optimal for other model organisms. The lack of overlap between the telomere interacting proteins in human cells identified in this study with in vivo proteomic studies gives weight to these caveats.

If the manuscript is substantially rewritten to include the caveats of this approach and focuses on the shelterin proteins then it is suitable for publication.

Author point-by-point response

Reviewer: black

Authors: blue

Reviewer #3 (Remarks to the Author):

Phylointeractomics reconstructs functional evolution of protein binding

Kappie, et al.

The authors have addressed most of the concerns raised in my original review. The *in vitro* binding data with the chicken TERF1 BD is a good addition. The new data makes a stronger case for the interesting hypothesis about the evolution of shelterin-telomere interaction. However, the main criticism is still relevant. The authors claim that the complexes that they are building on telomeric sequence *in vitro* necessarily reflect the *in vivo* arrangements. The abstract states, “To investigate such functional evolution, we here combine phylogenomics with interaction genomics.” I cannot find a description of functional genomics, which includes analysis of protein interactions that are built *in vitro* from a lysate on a substrate. For example, the review article about functional genomics (PMID:17593931) that is cited by the authors only describes functional genomics as interactions that are observed in complexes that are formed in the cell. This is not a minor distinction. Immuno-precipitation of complexes combined with MS is a powerful method that determines relationships between proteins in the cell. Adding a lysate to build complexes *in vitro* may lose many of the *in vivo* relationships and interactions that do not usually form due to temporal or spatial differences. Further it could amplify weak interactions or potentially create interactions that do not naturally occur *in vivo*. PMID:17593931 states post translational modifications determine complex formation

There might be a misunderstanding as we do not speak of “functional genomics” nor “interaction genomics” in the abstract, but of “interaction proteomics”. We do not build the complex as done in crystallization experiments for example, but we catch posttranslationally modified proteins and pre-formed complexes with our bait similar to wide-spread protein-protein interaction studies. The applications of interactomics have been strongly extended in the last 10 years beyond immuno-precipitations as evident from the publications which used bait and lysate in different instances (histone binding proteins¹⁻⁵, chemically synthesized nucleosomes^{2,8}, DNA⁹⁻²² and RNA²³⁻²⁵). The cited review may have not been the best choice to review these specialized applications (as written in 2007) and we have exchanged it against a more recent and focused review (Proteomics to study DNA-bound and chromatin-associated gene regulatory complexes, PMID: 27402878). To instruct the reader on potential confounding factors, we added a complete section to the manuscript. We would like to highlight that TERF1 has originally been identified through the same experimental strategy: protein enrichment on bait DNA followed by protein identification with mass spectrometry (Chong et al., Science 1995; PMID: 7502076).

1. Vermeulen, M. *et al.* Quantitative interaction proteomics and genome-wide profiling of epigenetic histone marks and their readers. *Cell* **142**, 967–980 (2010).
2. Nikolov, M. *et al.* Chromatin affinity purification and quantitative mass spectrometry defining the interactome of histone modification patterns. *Mol Cell Proteomics* **10**, M110.005371 (2011).
3. Eberl, H. C., Spruijt, C. G., Kelstrup, C. D., Vermeulen, M. & Mann, M. A map of general and specialized chromatin readers in mouse tissues generated by label-free interaction proteomics. *Mol Cell* **49**, 368–378 (2013).
4. Bluhm, A., Casas-Vila, N., Scheibe, M. & Butter, F. Reader interactome of epigenetic histone marks in birds. *Proteomics* **16**, 427–436 (2016).
5. Oda, H. *et al.* Regulation of the histone H4 monomethylase PR-Set7 by CRL4(Cdt2)-mediated PCNA-dependent degradation during DNA damage. *Mol Cell* **40**, 364–376 (2010).
6. Wysocka, J. *et al.* A PHD finger of NURF couples histone H3 lysine 4 trimethylation with chromatin remodelling. *Nature* **442**, 86–90 (2006).
7. Shema-Yaacoby, E. *et al.* Systematic identification of proteins binding to chromatin-embedded ubiquitylated H2B reveals recruitment of SWI/SNF to regulate transcription. *Cell Rep* **4**, 601–608 (2013).
8. Bartke, T. *et al.* Nucleosome-interacting proteins regulated by DNA and histone methylation. *Cell* **143**, 470–484 (2010).
9. Spruijt, C. G. *et al.* Dynamic readers for 5-(hydroxy)methylcytosine and its oxidized derivatives. *Cell* **152**, 1146–1159 (2013).
10. Kappei, D. *et al.* HOT1 is a mammalian direct telomere repeat-binding protein contributing to telomerase recruitment. *EMBO J* **32**, 1681–1701 (2013).
11. Casas-Vila, N., Scheibe, M., Freiwald, A., Kappei, D. & Butter, F. Identification of TTAGGG-binding proteins in *Neurospora crassa*, a fungus with vertebrate-like telomere repeats. *BMC Genomics* **16**, 965 (2015).
12. Butter, F., Kappei, D., Buchholz, F., Vermeulen, M. & Mann, M. A domesticated transposon mediates the effects of a single-nucleotide polymorphism responsible for enhanced muscle growth. *EMBO Rep* **11**, 305–311 (2010).
13. Butter, F. *et al.* Proteome-wide analysis of disease-associated SNPs that show allele-specific transcription factor binding. *PLoS Genet* **8**, e1002982 (2012).
14. Powers, N. R. *et al.* Alleles of a polymorphic ETV6 binding site in DCDC2 confer risk of reading and language impairment. *Am J Hum Genet* **93**, 19–28 (2013).
15. Viturawong, T., Meissner, F., Butter, F. & Mann, M. A DNA-centric protein interaction map of ultraconserved elements reveals contribution of transcription factor binding hubs to conservation. *Cell Rep* **5**, 531–545 (2013).
16. Mittler, G., Butter, F. & Mann, M. A SILAC-based DNA protein interaction screen that identifies candidate binding proteins to functional DNA elements. *Genome Res* **19**, 284–293 (2009).
17. Hubner, N. C., Nguyen, L. N., Hornig, N. C. & Stunnenberg, H. G. A quantitative proteomics tool to identify DNA-protein interactions in primary cells or blood. *J Proteome Res* **14**, 1315–1329 (2015).
18. Makowski, M. M. *et al.* An interaction proteomics survey of transcription factor binding at recurrent TERT promoter mutations. *Proteomics* **16**, 417–426 (2016).
19. Trung, N. T., Kremmer, E. & Mittler, G. Biochemical and cellular characterization of transcription factors binding to the hyperconserved core promoter-associated M4 motif. *BMC Genomics* **17**, 693 (2016).
20. Himeda, C. L. *et al.* Quantitative proteomic identification of six4 as the trex-binding factor in the muscle creatine kinase enhancer. *Mol Cell Biol* **24**, 2132–2143 (2004).
21. Himeda, C. L., Ranish, J. A. & Hauschka, S. D. Quantitative proteomic identification of MAZ as a transcriptional regulator of muscle-specific genes in skeletal and cardiac myocytes. *Mol Cell Biol* **28**, 6521–6535 (2008).
22. Ranish, J. A. *et al.* Identification of TFB5, a new component of general transcription and DNA repair factor IIH. *Nat Genet* **36**, 707–713 (2004).
23. Butter, F., Scheibe, M., Morl, M. & Mann, M. Unbiased RNA-protein interaction screen by quantitative proteomics. *Proc Natl Acad Sci U S A* **106**, 10626–10631 (2009).
24. Scheibe, M., Butter, F., Hafner, M., Tuschl, T. & Mann, M. Quantitative mass spectrometry and PAR-CLIP to identify RNA-protein interactions. *Nucleic Acids Res* **40**, 9897–9902 (2012).
25. Scheibe, M. *et al.* Quantitative interaction screen of telomeric repeat-containing RNA reveals novel TERRA regulators. *Genome Res* **23**, 2149–2157 (2013).

The cleared extract generated using the Dignam et al., 1983 nuclear extract protocol could concentrate free telomere proteins that are modified to not interact with the telomere as the chromosomes are spun out. Therefore, it is possible that the complexes built and interactions formed in vitro do not reflect the interactions that occur in the cell and are not “unbiased”, as claimed in the manuscript.

The nuclear protein extraction of the Dignam method is based on a hypertonic step with 420 mM NaCl that also dissociates and extracts a fraction of chromatin-bound proteins as this salt concentration exceeds physiological levels. Furthermore, given the improvements in interactomics, different techniques (PiCh, ChIP-MS, mChIP, RIME and even IP-MS) can be used to gain additional insight. This discussion has been added to the manuscript and the word “unbiased” was removed.

Furthermore, the lack of G-tail in the substrates used to build shelterin from the different species could bias against organisms that may be dependent on POT1 binding for the association of the other shelterin and telomeric proteins. Finally, the Dignam et al protocol was developed for human cells and may not be optimal for other model organisms.

We did not build shelterin, but used TTAGGG repeat oligonucleotides as bait. Our experimental setup is not more prone to confounding factors than any other study using similar techniques (see current publications above). As judged from the overlap of detected non-enriched proteins by mass spectrometry, the Dignam method showed similar extraction performance for cultured cells from the different vertebrates.

The lack of overlap between the telomere interacting proteins in human cells identified in this study with in vivo proteomic studies gives weight to these caveats.

We want to point out that the human *in vivo* proteomics studies are also not optimal and suffer from various caveats. This is evident from the comparison between different studies that had been requested by the reviewer during the previous submission round: the overlap between *in vivo* studies themselves is small (basically the shelterin complex) despite candidate lists with hundreds of proteins (**Supplementary Fig. 3**).

If the manuscript is substantially rewritten to include the caveats of this approach and focuses on the shelterin proteins then it is suitable for publication.

We have now included a full paragraph about the caveats and alternative strategies. As discussed with the editor, we did not put more focus on the shelterin proteins as we only use telomeres to exemplify the concept of phylointeractomics.